# Experimental Study on Two-Dimensional Rotatory Ultrasonic Combined Electrochemical Generating Machining of Ceramic-Reinforced Metal Matrix Materials

**DOI:** 10.3390/s22030877

**Published:** 2022-01-24

**Authors:** Wanwan Chen, Jing Li, Yongwei Zhu

**Affiliations:** 1College of Mechanical Engineering, Yangzhou University, Yangzhou 225127, China; hurrypipi@163.com (W.C.); ljing724@163.com (J.L.); 2College of Hydraulic Science and Engineering, Yangzhou University, Yangzhou 225127, China

**Keywords:** combined electrical machining, 2D ultrasonic, machining experiments

## Abstract

According to the machining characteristics of ceramic-particle-reinforced metal matrix composites, an experimental study on difficult-to-machine materials was carried out by two-dimensional (2D) rotatory ultrasonic combined electrolytic generating machining (RUCEGM), which organically combined an ultrasonic effect with a high-speed rotating tool electrode and electrolysis. After building the one-dimensional (1D) and 2D-RUCEGM systems, the factors influencing the combined machining process were analyzed and the experiments on RUCEGM were conducted to explore the feasibility and advantages of 2D-RUCEGM. The experimental results showed that, compared with 1D-RUCEGM, 2D-RUCEGM had higher accuracy, which increased about 21% and also reduced the machining time. Under certain conditions, the efficiency of 2D-RUCEGM was proportional to the voltage, and the machining efficiency could be enhanced by increasing the feed rate. The inter-electrode voltage detection module used in the experiment could improve the machining stability of the system.

## 1. Introduction

Due to their high strength, high hardness, and excellent wear resistance, ceramic-grain-reinforced metal matrix composites can be used for monolithic metal materials or traditional alloy materials in many specific applications, including aviation, aerospace, transportation, and others, and they are being gradually industrialized, with important applications in the design and manufacture of new components [1,2]. Nowadays, the main problems of metal matrix composites in traditional machining are shown as follows: (1) increased machining costs caused by severe tool wear during machining; (2) difficulty in ensuring the integrity of the machined surface for the breakage, shedding, and scratching of SiC particles on the machined surface; (3) failure to meet the technical requirements of dimensional accuracy and stability of parts for the uneven stress distribution in the material; and (4) low machining efficiency and long machining cycle caused by the small machining parameters of traditional cutting and grinding methods [3]. Ultrasonic vibration is used to assist electro-machining and remove the electrolytic product (anode passivation film) by impact scraping of abrasive particles. In the process of electrical discharge machining, ultrasonic cavitation and pumping can be used to effectively remove electric corrosion in real time, speed up the circulation of the working fluid, improve the gap discharge condition, effectively avoid arc discharge, increase the effective pulse ratio, and increase the depth–diameter ratio of the machined hole, besides spark discharge [4]. The organic combination of ultrasonic vibration, spark discharge, and electrolysis not only keep the technical advantages of every single process, but can also abandon its independent technical limitations. Due to its advantages of not being affected by material hardness, small machining force, and low tool loss, this is an effective method to solve the machining and manufacturing of such special-performance composite materials [5,6,7].

Through rotatory ultrasonic milling, micro-channels with a good surface finish (Ra = 0.21 μm), smooth morphology, and minimum edge fragmentation (16.3 μm) were milled on alumina ceramics and, by selecting appropriate rotatory ultrasonic machining (RUM) parameters, Basem [8] explored the machinability of alumina ceramics. By applying the principle and characteristics of RUM to remove hard and brittle materials, Kuo [9] found the good effect of ultrasonic-assisted milling on reducing machining resistance in the milling process, and its faster milling speed to remove materials. Based on the hydrodynamic method, S. Skoczypiec [10] analyzed the electrolyte flow between the cathode and anode in ultrasonic combined electrochemical machining, and verified via experiments that ultrasonic vibration could improve the electrolyte flow in the electrolytic gap. To improve the grinding performance of titanium alloys, Japanese scholars Li S, et al. [11] found that ultrasonic vibration promoted the plasma electrolytic oxidation of the workpiece surface and reduced the hardness, the grinding force, and the roughness of the workpiece surface under the condition of plasma electrolytic oxidation.

Wang, et al. studied the application of rotatory ultrasonic in machining CFRP materials under different aspects and established a material removal model. The experimental results showed that, although the cutting force decreased significantly, the surface roughness also increased under the condition of ultrasonic combination [12,13,14,15]. Lu, et al. designed a new three-dimensional ultrasonic vibration platform with tunable characteristics, modeled it, and verified its functioning via experiments. The platform could realize a variety of vibration modes, including 1D (linear) and 2D (in-plane or out-of-plane) vibrations [16]. The properties of composite materials related to Al_2_O_3_ have been studied by A.M. Sadoun, et al. [17,18,19].

Zhu Yongwei, et al. developed an ultrasonic synchronous combined microfabrication system and carried out a series of basic research and application experiments, which verified the technical advantages of the combined machining method [20,21,22,23].

Based on rotatory ultrasonic combined electrochemical machining, this paper compares 1D and 2D RUCEGM technology. In particular, we stimulated and enhanced the machining process between electrolysis and discharge electrodes using the rotating ultrasonic vibration effect of the tool electrode to realize real-time and synchronous control of the “on/off” of the electric machining power supply. Further, we explored the influence of related parameters on the machining system during the combined machining process to find the technical advantages of multi-effect synergy, such as synchronous ultrasonic vibration, small gap pulse electrolysis, micro-electrical discharge, etc. Thus, this paper provides a new research idea for micro-precision and high-efficiency machining of difficult-to-machine materials and special-shaped parts.

## 2. Design of 2D-RUCEGM System

### 2.1. Fundamentals of RUCEGM

In the process of the RUCEGM, metal materials are mainly used for corrosion removal by pulse electrolysis and plastic removal of abrasive grains (mechanical grinding) under rotatory vibrations, accompanied by a small amount of electric spark discharge corrosion of materials. The electrolytic hydrogen bubbles, namely, the cavitation bubbles generated in the gap and gathering on the surface of the tool, reduce the conductivity of the electrolyte. With the decrease in the gap, the electrolyte is broken down at the maximum point of field strength and forms a tiny spark discharge, which helps to remove local difficult-to-process points. Therefore, to give full play to the enhanced mass transfer effect and coupling modulation effect of ultrasonic, a new 2D vibration direction (main vibration direction and auxiliary vibration direction) was obtained via an online detection and control system to add and adjust a radial ultrasonic (*x*-direction or *y*-direction) vibration device to the workbench based on traditional 1D-RUCEGM. The fundamentals of RUCEGM are shown in Figure 1.

A diamond-consolidated mill grain tool is used as a tool electrode to make the workpiece with a certain micro-pressure. The positive electrode of the pulse power supply is connected to the workpiece and the negative electrode is connected to the tool electrode. Electrolytes are added to the machining zone. During the machining process, the diamond-consolidated mill grain tool rotates axially while generating ultrasonic vibrations. When the workpiece generates ultrasonic vibrations in the radial direction, there is a minimum gap L_min_ between the tool electrode and the workpiece with a size about the diameter of the grains on the end face of the tool. According to the motion characteristics of the tool in rotatory ultrasonic machining, the motion trajectory of a single diamond grain on the end face of the tool in the *x*-, *y*-, and *z*-directions can be described by the following formula:(1)$$\begin{array}{c} x\left(t\right)\\ y\left(t\right)\\ z\left(t\right) \end{array}=\{\begin{array}{c} v_{s}t+r\cos\left(\frac{\pi n}{30}t\right)+A_{x}\sin\left(2\pi f_{x}t\right)\\ r\sin\left(\frac{\pi n}{30}t\right)\\ A_{z}\cos\left(2\pi f_{z}t\right)+z_{0} \end{array}$$
where $v_{s}$ refers to the feed speed; r refers to the tool radius; n refers to the axial rotating speed; $A_{x}$ and $A_{z}$ refer to the triaxial ultrasonic amplitude during machining; $f_{x}$ and $f_{z}$ refer to the frequency of ultrasonic vibration corresponding to each axis; *t* refers to the time. As at the initial end face gap, $z_{0}$ is a constant value. During rotatory ultrasonic plane machining, the feed rate of the diamond grains in the *z*-direction is 0 and there is a feed rate in the *x*-direction or *y*-direction.

According to the trajectory, different from the motion trajectory of diamond grains on the end face of tools in common machining, the motion characteristics of diamond grains are changed by ultrasonic vibration in RUM. The diamond grains no longer perform only a single spiral motion, but there is a complex three-dimensional motion in a space. Therefore, the relative position between the diamond grains and the workpiece is changing at all times. The motion equation of the abrasive particles is established according to the motion track of the tools, the in-unit feed period, and rotation period. The material removal law of RUM can be analyzed, and the tool feed rate is calculated as follows:(2)$$v_{z}\geq \Delta k\delta n$$
where δ is the removal depth of comprehensive materials; Δk refers to the ratio of the actual removal material’s value and its value at the theoretical depth of δ, indicating the efficiency of machining completion in the rotation period, which is related to the feed rate, grinding force, and material properties. Compared the machining without ultrasonic vibration, the effective machining gap of rotatory ultrasonic combined electrochemical machining on the micro-hole bottom is smaller. δn=LM, that is, the turnover removal volume number of RUM is related to the rotation rate of the tools and the maximum grinding depth of diamonds. According to our analyses, the bigger the value of δn is, the smaller the grain size and the electrochemical machining gap and the higher the machining efficiency.

Due to the effect of rotatory ultrasonic vibrations, the machining gap between the tool end face and the micro-hole bottom surface changes periodically. The smaller the value is, the higher the current density is, the more metal-based materials are eroded, and the greater the proportion of exposed ceramic particles is. For RUM, the bigger the machining depth is, the higher the machining efficiency will be. Therefore, the current density in pulse electrochemical machining varies from time to time. Taking the *z*-axis as the example, the rate of the axial electrochemical machining at *t* can be calculated as:(3)$$v_{a}=D\eta \omega i=D\eta \omega \frac{\sigma U_{R}}{z(t)}$$

The actual depth of electrochemical machining can be shown as:(4)$$h_{z}=D\eta \omega \int_{t_{1}}^{t_{2}}i(t)dt=D\eta \omega \int_{t_{1}}^{t_{2}}\frac{\sigma U_{R}}{z_{0}+\sin(2\pi ft)}dt$$
where $v_{a}$ refers to the etching rate, $z_{0}$ refers to the initial gap, $U_{R}$ refers to the voltage drop in interstitial electrolyte, D refers to the duty cycle of pulse power supply, σ refers to the electrical conductivity of the electrolyte, η refers to the current efficiency, and ω refers to the electrolytic equivalent volume of the substance electrolyzed.

The ultrasonic vibration can effectively eliminate the short circuit of inter-electrode discharge. Rotatory axial ultrasonic vibrations can rate up the circulation renewal of the working fluid and the removal of machined products, as well as promote the uniformity of the flow field and electric field, and enhance the local removal ability of materials.

In actual machining, the surface margin distribution of the workpiece after etching of the material is not uniform and the machining gap is also uneven [24]. The rotatory ultrasonic vibration effect in the electrochemical machining area reduces the gap of electrochemical machining. For the metal protrusion part, the etching speed is quicker than that of the low concave part and the etched material is more abundant. The metal protrusion part is flattened to improve the metal-forming flatness of the machining bottom surface and increase the exposed ratio of ceramic grains, as shown in Figure 2. This avoids metal adhesion to tool grains during grinding, thus improving the effectiveness of rotatory ultrasonic grinding of the micro-hole bottom surface and ensuring the machining quality. However, in the process of wall forming, electrolysis only erodes the metal-based materials on the wall; under the influence of the ultrasonic, some grains fall off to increase the surface roughness of the wall. With the increase in machining depth, the surface roughness near the entrance of the wall increases, which reduces the dimensional forming accuracy.

When the rotating rate, amplitude, and grinding material properties of RUM are changed, the material removal rate and surface roughness of RUM change in each process. Therefore, appropriate matching parameters need to be chosen to reasonably distribute the corresponding proportion of the machining process and maximize the efficiency of the combined machining system. By adjusting the ultrasonic power, the ultrasonic amplitude of the tools and workpieces can be changed. The larger the amplitude is, the higher the machining efficiency; with the increase in the rotating speed of the main axis, the machining efficiency becomes higher and the surface margin becomes smaller. When the grain radius (granularity) increases, the material removal speed increases, but the machined surface may be rough. Electrolysis can be mainly changed by adjusting the pulse voltage amplitude, pulse ratio, and electrolyte conductivity. When the voltage, pulse ratio, and electrolyte conductivity become larger, the electrolysis energy increases and the machining efficiency becomes higher. However, in this case, the stray etching phenomenon becomes severe and the surface quality of the workpiece is unacceptable.

### 2.2. Analysis of the Coupling Relationship between Ultrasonic Parameters and Electrolytic Parameters in Machining Process

In the machining process, a new 2D vibration direction was synthesized by online detecting and adjusting the output power of the ultrasonic generator. The main vibration direction was kept consistent with the feeding direction of the cathode tool. Through the high-frequency vibration and its cavitation effect, the removal efficiency of electrolyte products and the renewal speed of the electrolytes in the gap, as well as the flow field in the machining area, were improved to realize electrolytic milling machining with a smaller gap. Meanwhile, the coupled main vibration was also designed to play the important role of adjusting the co-ordination between high-voltage discharge and low-voltage electrolysis. Through the computer control system, the coupling relationship between ultrasonic parameters and variable voltage parameters could be accurately adjusted, and high-voltage discharge could be carried out in the small gap area to eliminate the dangerous point area.

In the process of RUCEGM, the coupling relationship between ultrasonic parameters and electrolytic parameters is shown in Figure 3. The whole machining process can be divided into two stable machining states that continuously alternate, as explained below.
(1)The partial discharge state of ultrasonic modulation variable voltage, shown in section AB in Figure 3, is accompanied by high-speed electrolytic dissolution of the anode workpiece. During machining, when the coupling main vibration drives the cathode tool to approach the anode workpiece to the minimum machining gap, the power supply outputs high-frequency and high-voltage pulse electrical parameters ($U_{2}$, the discharge voltage in the figure) with the aim of forming a single or continuous forced spark discharge in the dangerous point area where bubbles are relatively concentrated in the machining gap; quickly removing materials in the dangerous point area; expanding the machining gap; and dispersing and removing the accumulated bubbles and electric corrosion products via the high-speed flowing electrolyte from the machining area.(2)The ultrasonic-assisted high-efficiency electrochemical machining state, as shown in the BCA section. In this state process, the power supply outputs constant voltage electrical parameters ($U_{1}$, the electrolysis voltage in the figure), accompanied by ultrasonic high-frequency vibrations. The state can be subdivided into the two processes described below.

① The coupling main vibrations drive the cathode tool away from the anode workpiece, as shown in the BC section in Figure 3. Because the discharge eliminates the dangerous point area in the gap, this state can be implemented smoothly. Meanwhile, the gap between electrodes is relatively large in the process and the electrolytes flowing at high speed quickly remove from the machine area the electrolytic products produced by electrolytic dissolution, the electric corrosion products produced by the AB section discharge, and the residual heat away from the machining area, thus avoiding uncontrollable discharge burns with high energy density and ensuring a smooth machining process.

② The coupling main vibration drives the cathode tool away from the anode workpiece, as shown in the CA section in the figure. During this process, the gap between electrodes shrinks rapidly and the flow field in the gap deteriorates sharply. Electrolytic products accumulate rapidly in the distorted flow field area in the gap, the conductivity decreases continuously, the side reactions increase, and a large number of bubbles are produced and accumulate gradually to form a dangerous point area that may induce a short circuit.

Further analysis shows that, due to the vibration produced by the ultrasonic, the electrode gap fluctuated between the maximum gap Δ_max_ and the minimum gap Δ_min_. Generally, when the electrode gap reaches the minimum gap value range, the machining short circuit will occur or has occurred. Therefore, the amplitude of the voltage difference between the two electrodes needs to be adjusted and the output voltage of the power supply needs to be changed from the electrolytic voltage $U_{1}$ to the discharge voltage $U_{2}$. Consequently, in the gap, the insulating medium in the local dangerous point area is broken down under the action of discharge voltage, forming instantaneous cremation discharge, at which time the material in the dangerous point area is rapidly melted and thrown out by electricity.

To ensure the smooth implementation and switches of the above machining process, it was necessary to build a multi-energy field synergistic control system. Through the acquisition of machining process parameters, the electromechanical parameters of the machining system can be detected and regulated online, and the accurate coupling of ultrasonic parameters and variable voltage parameters can be realized.

### 2.3. Composition of the Machining System

To realize the 2D-RUCEGM, it is necessary to combine the vibration in two directions with the same frequency and phase, that is, to synthesize the vibration in the same phase and perpendicular to the workpiece. The experimental system was built and the synchronous controller is designed to transmit the ultrasonic signal to the ultrasonic generator in the *x*/*y*-direction at the same time; the synthetic straight line coincided with the normal direction of the theoretical generating curved surface. When the generating curved surface is machined, the tool head vibrates along the generating generatrix direction and the synthetic motion vibrates along with the generating normal direction.

The experimental system consists of a displacement movement platform, a rotatory ultrasonic vibration system, radial ultrasonic vibration system, an electrochemical machining system, and a control system. The workpiece driven by the moving platform can realize the linear motions of *x*, *y*, and *z*. The axial rotation ultrasonic vibration system is composed of the ultrasonic transducer, horn, and tool head, which makes the tool generate high-frequency ultrasonic vibration while rotating at high speed. The radial ultrasonic vibration system is mainly composed of two piezoelectric ceramic transducers with a diameter of 50 mm, which can generate vibrations with a diameter of 50 mm and an amplitude of 3~7 mm by voltage regulation. Therefore, for such a workpiece, ultrasonic vibrations can be carried out in the radial direction (the *x-* or *y*-axis) or, alternatively, rotatory ultrasonic vibration can be carried out in the direction of the *z*-axis to realize the 2D-RUCEGM. The pulse electrolytic power supply provides the electric energy of electrochemical machining needed in the combined machining process, and the control subsystems are unified and combined into a complete system to achieve the machining goal.

The ultrasonic vibration system in the 2D-RUCEGM system can work simultaneously in a single pair or single selection to transmit the vibration energy to the workpiece and realize its 2D ultrasonic vibration machining. The motion related to the ultrasonic vibration in 1D- and 2D-RUCEGM is explained in detail below:(1)When there is only ultrasonic vibration of the rotating axis (*z*-direction), that is, when the side surface is machined by 1D ultrasonic vibration, the tool and the workpiece are always in contact and the machining gap in combined electrochemical machining is difficult to utilize; therefore, only the end face of the tool can be selected for replication-forming machining.(2)When there is synchronous ultrasonic vibration along the *z*- and *x*-axes or the *z*- and *y*-axes, the vibrations in the machining direction of the side surface of the 2D ultrasonic vibration tool ensure the electrolysis action in the machining gap, and the generating machining can only be conducted on the side surface of the tool.

According to the above analysis, the RUCEGM system, whose schematic diagram is shown in Figure 4, was preliminarily designed. The workbench was a multi-axis linkage control feed mechanism and the servo motor is driven by a servo driver to realize micro-feed movement in multi-dimensional space, thus ensuring the stability of the machining process. The axial ultrasonic generator would be connected to the rotating ultrasonic principal axis and the radial ultrasonic generating system was fixed on the workbench to generate radial vibrations. The positive electrode of the electrolytic power supply was connected to the workpiece and the negative electrode was connected to the tool head. The electrolyte is injected by hydrostatic injection in the machining process, so that the electrolysis could be realized.

## 3. Experimental Analyses

According to the machining system described in the second chapter, a 2D-RUCEGM system was built. Contrast experiments were carried out on grain-reinforced ceramic materials and the experimental results are here reported.

### 3.1. Experiment Equipments

The machining system included the following: a workbench, fixture, ultrasonic vibration system, micro-displacement sensor, signal amplifier, data acquisition card, synchronous controller, control unit, inter-electrode voltage detection device, etc. When the system was machined, the *z*-direction vibration and feed of the rotatory ultrasonic tool head, the radial ultrasonic feed motion, and the normal direction of the coupled vibration surface were applied to the workpiece. When the tool head ground the surface of the workpiece, due to the normal vibration of the workpiece, the grinding force periodically acted on the workpiece in the form of a pulse, and the electrochemical machining also acts on the workpiece with pulse voltage, which allowed combined machining to reach the optimal state.

Due to the minimum value phenomenon of the gap between electrodes in the combined machining process, generally a short-circuit protection avoidance method based on the detection of inter-electrode voltage is adopted in electrochemical machining. This method can be used to evaluate whether there is a short circuit thanks to the phenomenon whereby the inter-pulse voltage is not zero during the process of pulse power supply electrochemical machining. If the inter-pulse voltage is close to zero, indicating the electric spark discharge, it is necessary to control the output voltage of the pulse power supply by feedback signal in time and modulate the ultrasonic amplitude synchronously. At the same time, the servo system of the machine tool controls the recoil of the tool electrode to eliminate the electric spark phenomenon in a short time. The composition of the detection module is shown in Figure 5.

By collecting the inter-pulse voltage between the tool electrode and the workpiece, the inter-electrode voltages are introduced into the comparison judge device and detected, compared, and judged one by one, and the judgment signal is fed back to the pulse power supply system, the ultrasonic generation system, and the machine tool servo system to control the output of high and low potentials of the pulse power supply and the feed and back of the tool electrode.

### 3.2. Experimental Process and Analyses

From the above mechanism analysis, it can be found that ultrasonic vibration has a great influence on the machining effect. Therefore, the axial and radial ultrasonic vibrations had to be measured and analyzed before the experiments. The machining parameters and their coupling relationship were monitored and adjusted by the self-designed control system for the online detection and adjustment of electromechanical parameters, so that their vibration frequency could be ensured to be the same in the machining system. The experimental results are shown in Figure 6 and the machining experimental set-up is shown in Figure 7.

#### 3.2.1. Influence of Different Machining Methods on Machining Quality

1D- and 2D-RUCEGM (grinding material added into the electrolyte) was carried out on ceramic-grain-reinforced materials and the results were compared. The experimental parameters are shown in the Table 1.

The morphology after the completion of the two machining methods can be seen in Figure 8.

From the experimental results, it could be seen that the 1D-RUCEGM has generated deep grinding crack, a smooth edge, and micro-pits; its machine time is about 3.4 min; and the surface roughness is 4.89 μm. In contrast, the 2D-RUCEGM machining produced a smooth groove surface and edge, its machine time is about 2.7 min, and the surface roughness is 3.85 μm, and other groups draw similar conclusions. In other words, the machining accuracy was increased by about 21% and the machining time was reduced by about 20%. The main reason is that the 2D ultrasonic vibrations allowed the electrolytes in the machining gap to update in time, strengthened the electrolysis effect, and further improved the stability of the flow field in the gap. This experiment verified the feasibility and advantages of 2D-RUCEGM.

#### 3.2.2. Influence of Pulse Voltage on Machining Effect

In the combined machining systems, the pulse voltage is significant for the machining efficiency of a 2D machining system. Therefore, it was necessary to study the influence of pulse voltage on the machining effect. Taking the machining time of a same-size workpiece as a time node, the machining effect was evaluated according to the statistical results of the specific machining time and the surface roughness after machining.

The relevant experimental parameters are shown in Table 2.

The machining results are shown in Figure 9 and Figure 10. With the increase in machining voltage, the machining current density and the workpiece dissolution rate increased and the time for machining the same groove gradually decreased. However, with the increase in voltage value, the stray corrosion phenomenon became more severe and the machining quality decreased. In addition, the increase in the electrolytic etching energy per unit time enlarged the groove aperture. When the voltage reached 6 V, the phenomenon of spark discharge occurred in the machining process and the machining surface roughness decreased to 3.96 μm, even though the machining process was not interrupted by a short circuit under the action of the control system and the machine time reduced by 1.4 min. The reason for spark discharge is that, when the machining gap is increasingly small and the local point field strength is large enough, a discharge channel is formed after contacting the difficult-to-process protruding point. Meanwhile, the electrolytic hydrogen bubbles produced in the gap reduce the conductivity and produce spark discharge.

#### 3.2.3. Influence of Feed Rate on Machining Effect

It is an everlasting goal to improve the efficiency of the machining process. Through the above analysis, it was found that the short-circuit detection of inter-pulse voltage ensured the continuity of the machining. To verify the influence of feed rate on the machining effect, experiments on the machining of ceramic grain reinforced with the same size were designed with the implementation of different feed rates.

The relevant experimental parameters are shown in Table 3.

The machining results are shown in Figure 11 and Figure 12. With the increase in feed rate, the machining time, as well as the surface roughness value, first decreased and then increased. When the feed rate reached 4 mm/min, the machining time was the shortest, that is, the machining efficiency was the highest. When the feed rate was lower than 4 mm/min, the feed rate was relatively slow and failed to reach the optimal machining limit efficiency and the machining time is longer than that at 4 mm/min. However, when the feed rate was greater than 4 mm/min, the actual etching efficiency of machining exceeded the machining limit of the inherent conditions of the experiment; consequently, short circuits frequently appeared in the machining process and the number of short circuits increased with the increase in feed rate, which led to an increase in the proportion of time used for short-circuit backtracking in machining, which, in turn, reduced the machining efficiency and increased the total machining time. The inter-pulse short-circuit protection could ensure the continuity of machining, but the workpiece material and tool electrode were constant and the optimal machining efficiency of combined machining, that is, the feed rate, had an optimal value. When the feed rate of the tool electrode was lower than this value, the machining efficiency could not reach the highest value, the machining process was relatively stable, and the possibility of a short circuit in the machining loop was very low. When the feed rate of the tool electrode was higher than this value, frequent rollbacks for the short-circuit detection of pulse voltage also affected the machining efficiency. For combined machining, when other factors were constant, there was an optimum feed rate. When the optimum feed rate was equal to the dissolution rate of metal, then the surface roughness was good; when the optimum feed rate was lower or higher than that value, the surface roughness value was increased. Therefore, the best feed rate could achieve a better machining effect when the machining time was short and the surface roughness was guaranteed at the same time.

Similarly, in electrochemical machining, electrolytes with different properties have an impact on machining efficiency. In the next experiment, a passive NaCl solution with the same concentration, instead of the passive NaNO_3_ used in the previous experiment, was further studied. All other conditions were completely consistent with the previous experiment. The machining results are shown in Figure 13 and Figure 14. Similar to that in the NaNO_3_ solution, when the feed rate reached 4 mm/min, the machining time was the shortest and, when the feed rate was the same, the total machining time in the NaCl solution was economized by about 2.5% and roughness was increased by about 3.3% compared to that of NaNO_3_. That is, the machining time and surface roughness could not be optimized at the same condition. So, it is necessary to consider which electrolyte is selected for machining different workpieces.

## 4. Conclusions

In this paper, 2D-RUCEGM technology is taken as the research object to establish the combined machining system and the material removal efficiency model of 2D-RUCEGM, as well as to analyze the influencing factors of machining accuracy and efficiency. On the designed platform, experiments were carried out. According to the experiment, rotatory ultrasonic generating electrochemical machining could improve the efficiency of difficult-to-machine materials and the surface accuracy. From the research content, the following conclusions were obtained:(1)By comparing the experiments on 1D- and 2D-RUCEGM, we found that the groove machined by 2D-RUCEGM showed higher precision and smooth plane and bottom surface and edge, accompanied by machining accuracy, which was increased by about 21%, and the machining time was reduced by about 20%.(2)When other experimental parameters were kept constant, the machining efficiency was proportional to the voltage, which is consistent with the theoretical analysis of material removal. However, with the increase in electrolytic voltage, the phenomenon of stray corrosion became more severe, the machining quality decreased, and the electrolytic etching energy per time unit increased. When the voltage surpassed the 6 V, the phenomenon of spark discharge occurred in the machining process and the machining accuracy decreased.(3)Under the conditions whereby other experimental parameters were kept constant, a faster feed rate under the same electrolyte condition improved the machining efficiency. When there was inter-pole voltage detection to evaluate short circuit, machining could still be completed stably and efficiently with reducing short circuits, even though the upper limit of feed rate was exceeded. However, if the feed rate continued to increase, although the machining can still be completed, the number of short-circuit regressions increased and the machining efficiency was gradually reduced; the optimal feed rate is 4 mm/min.(4)Different electrolyte conditions also have made a difference to the machining results. In this study, when comparing the machining results of NaCl and NaNO_3_, the machining time was reduced by about 2.5%, while the machining accuracy was decreased by 3.3% in the same concentration condition. Therefore, in the actual machining process, it is important to select the appropriate electrolyte to optimize the machining process in accordance with the specific needs of the workpiece.

## Figures and Tables

**Figure 1 sensors-22-00877-f001:**
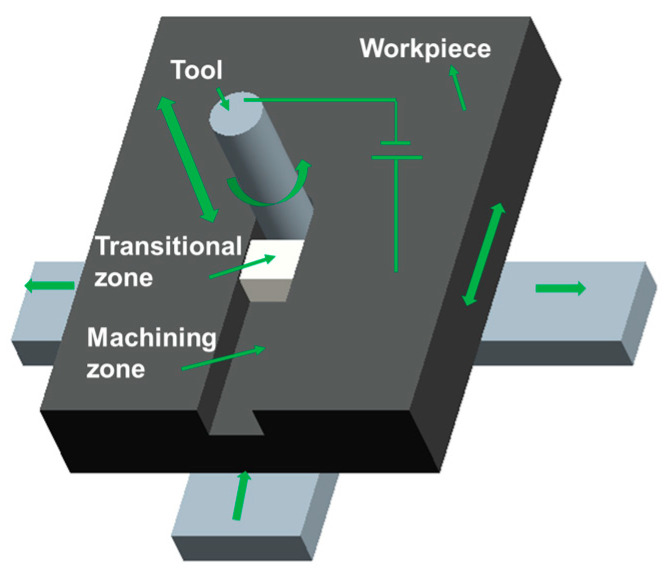
Mechanism of 2D-RUCEGM ultrasonic.

**Figure 2 sensors-22-00877-f002:**
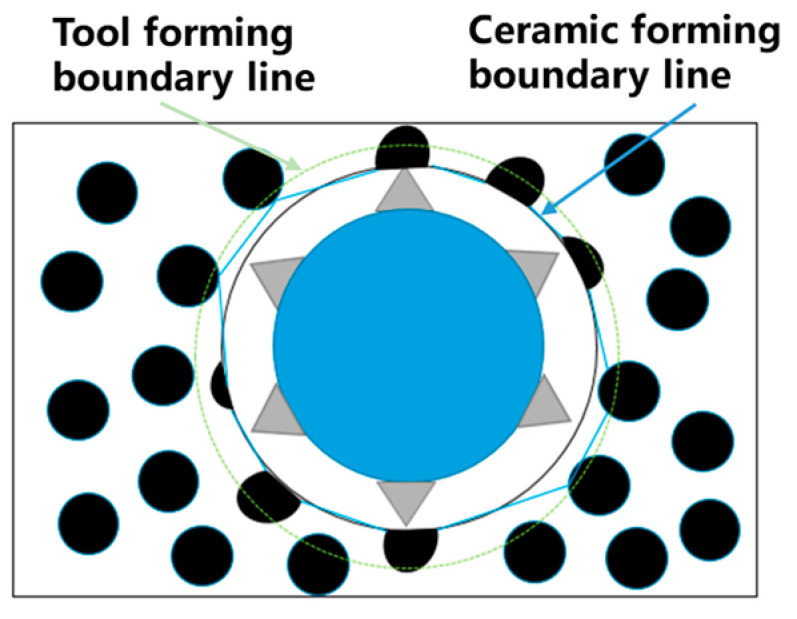
Tendency of microporous surface forming changes.

**Figure 3 sensors-22-00877-f003:**
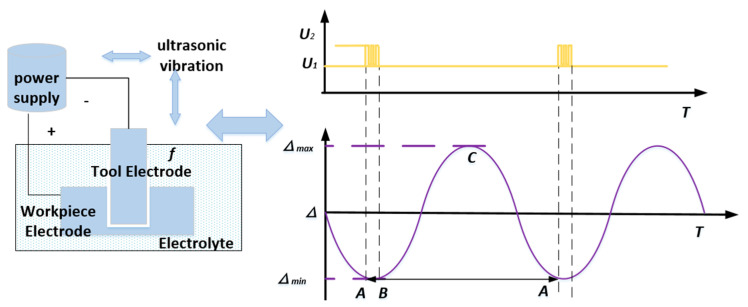
Schematic diagram of synergistic mechanism.

**Figure 4 sensors-22-00877-f004:**
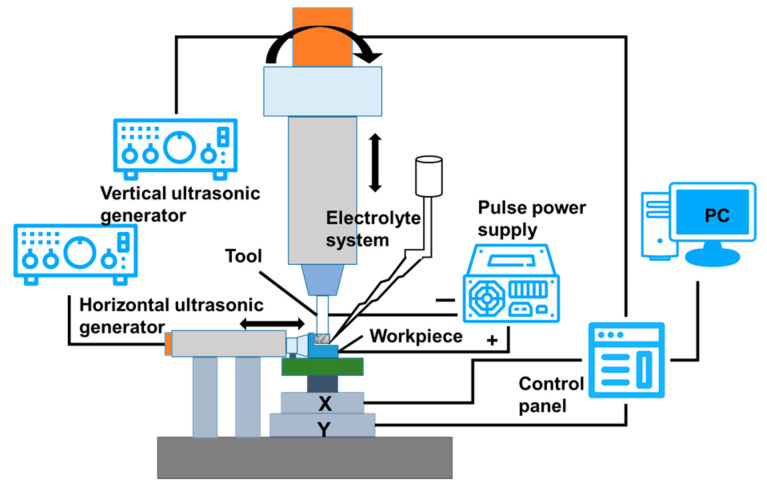
2D-RUCEGM system.

**Figure 5 sensors-22-00877-f005:**
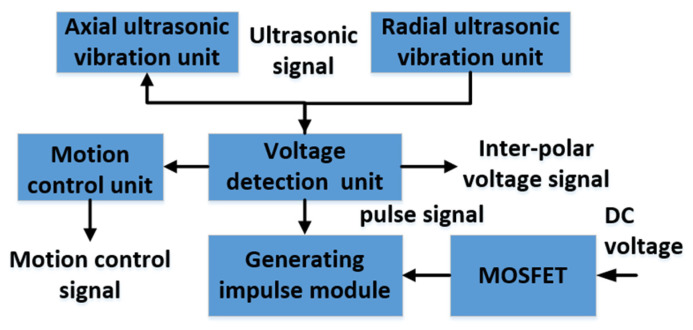
Detection system components.

**Figure 6 sensors-22-00877-f006:**
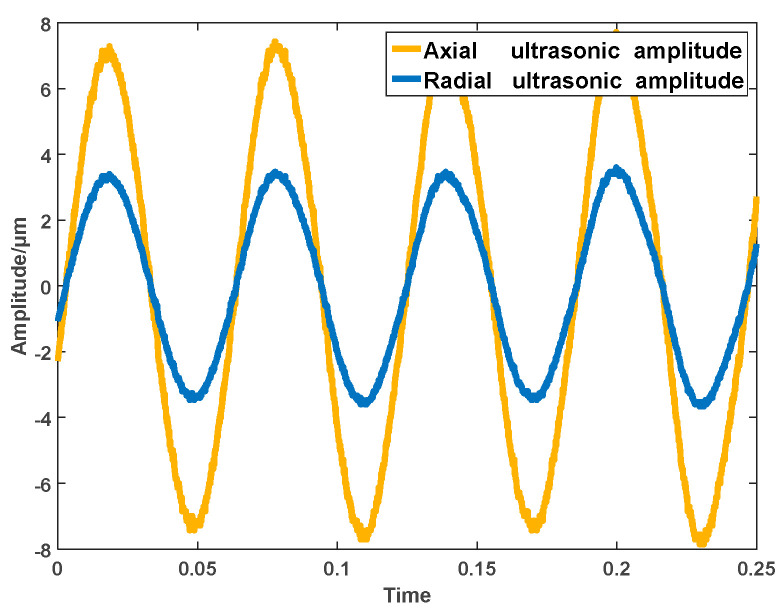
Axial and radial ultrasonic synchronization effects.

**Figure 7 sensors-22-00877-f007:**
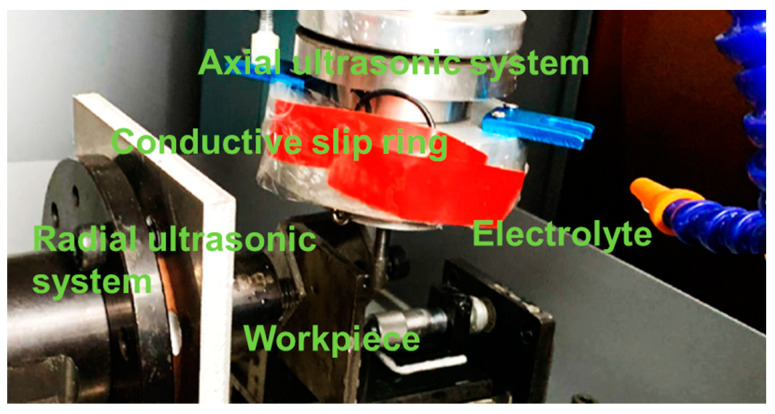
Partial machining experimental settings.

**Figure 8 sensors-22-00877-f008:**
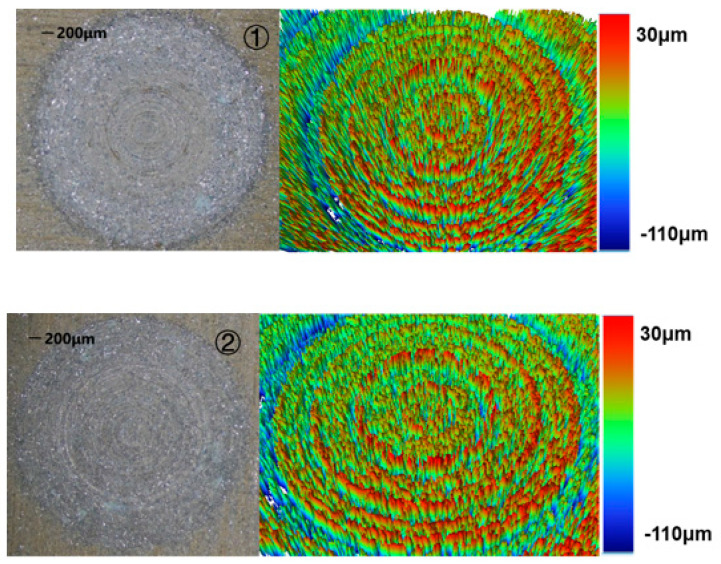
Features of two machining methods. ① Feature of the 1D machining; ② feature of the 2D machining.

**Figure 9 sensors-22-00877-f009:**
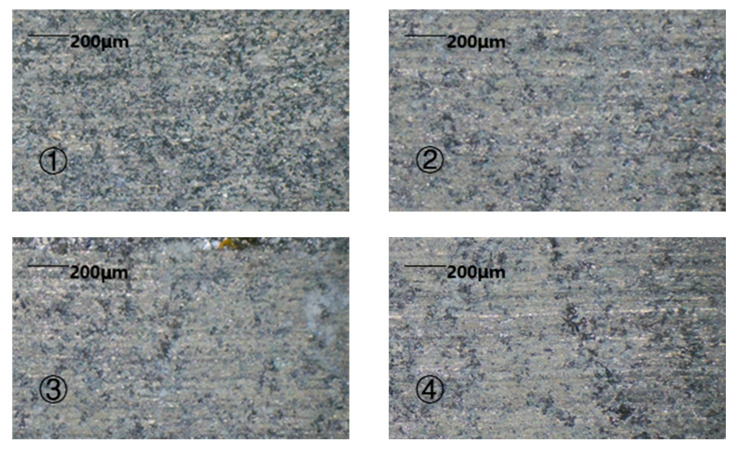
Resulting morphology for different voltage values: ① 3 V; ② 4 V; ③ 5 V; ④ 6 V.

**Figure 10 sensors-22-00877-f010:**
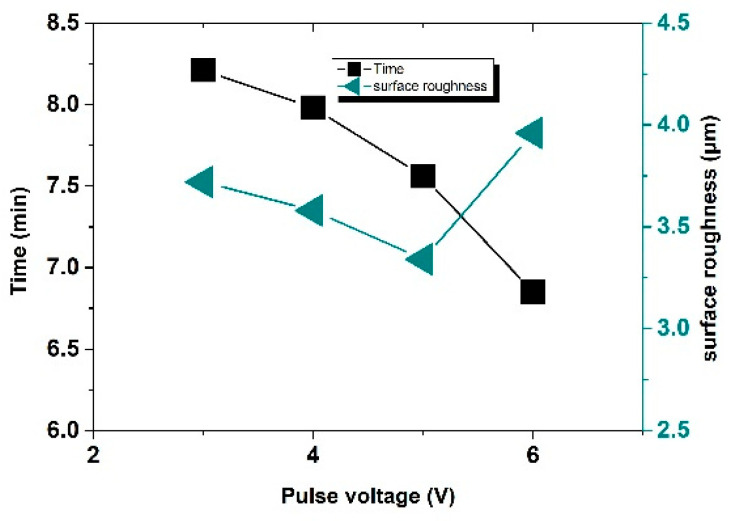
Roughness and machining time of different specimens under different voltages.

**Figure 11 sensors-22-00877-f011:**
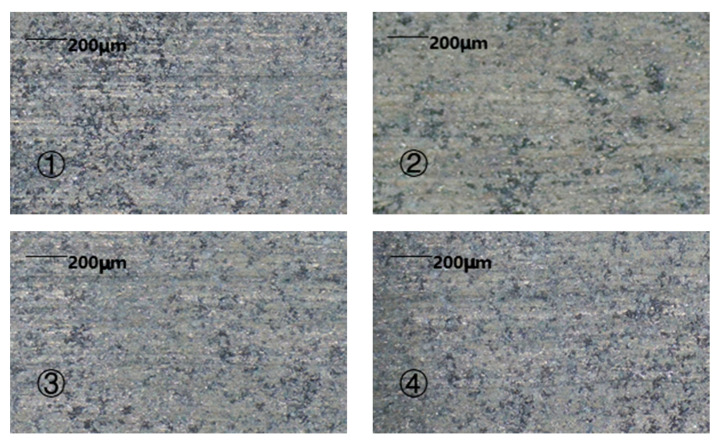
Resulting morphology for different feed rate values: ① 2 mm/min; ② 3 mm/min; ③ 4 mm/min; ④ 5 mm/min.

**Figure 12 sensors-22-00877-f012:**
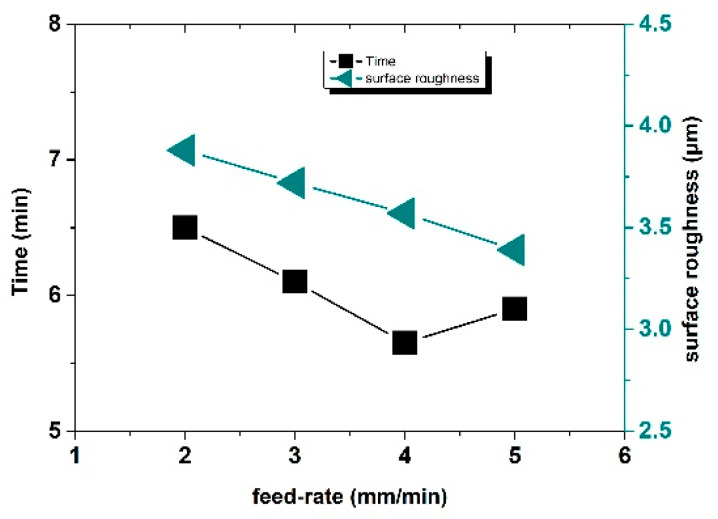
Roughness and machining time of different specimens under different feed rates.

**Figure 13 sensors-22-00877-f013:**
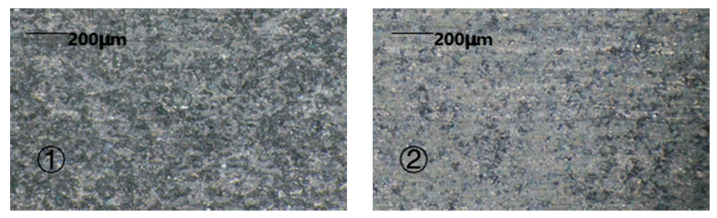
Resulting morphology with different solutions: ① NaCl, 4 mm/min; ② NaNO_3_, 4 mm/min.

**Figure 14 sensors-22-00877-f014:**
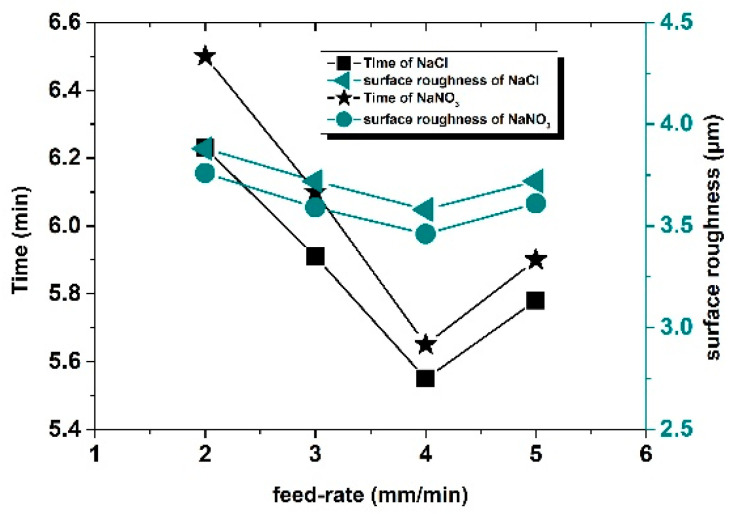
Roughness and machining time of feed rates in different electrolyte solutions on machining results.

**Table 1 sensors-22-00877-t001:** Material parameters.

Item	Characteristics
Workpiece	SiCp/Al composite, 65%; size, 30 mm × 30 mm × 8 mm.
Tool electrode	Diamond diameter: 6 mm; grain density, 100 mesh.
Electrolyte	NaNO_3_ mass fraction, 8%; grinding material, 400-mesh silicon carbide.
Electrolytic voltage	Duty cycle, 50%; voltage, 5 V.
Ultrasonic amplitude	Axial ultrasonic amplitude, 6 μm; radial ultrasonic amplitude, 3 μm.
Feed rate	5 mm/min.
Spindle speed	3000 r/min.

**Table 2 sensors-22-00877-t002:** Material parameters.

Item	Characteristics
Workpiece	SiCp/Al composite, 65%; size, 30 mm × 30 mm × 8 mm.
Tool electrode	Diamond diameter, 6mm; grain density, 100 mesh.
Electrolyte	NaNO_3_ mass fraction, 8%; grinding material, 400-mesh silicon carbide.
Pulse voltage	Duty cycle, 50%; voltage: 3 V, 4 V, 5 V, and 6 V.
Ultrasonic amplitude	Axial ultrasonic amplitude, 6 μm; radial ultrasonic amplitude, 3 μm.
Feed rate	4 mm/min.
Spindle speed	3000 r/min.

**Table 3 sensors-22-00877-t003:** Material parameters.

Item	Characteristics
Workpiece	SiCp/Al composite, 65%; size, 30 mm × 30 mm × 8 mm.
Tool electrode	Diamond diameter, 6 mm; grain density, 100 mesh.
Electrolyte	NaNO_3_ mass fraction, 8%; grinding material, 400-mesh silicon carbide.
Pulse voltage	Duty cycle, 50%; voltage, 5 V.
Ultrasonic amplitude	Axial ultrasonic amplitude, 6 μm; radial ultrasonic amplitude, 3 μm.
Feed rate	2 mm/min, 3 mm/min, 4 mm/min, 5 mm/min.
Spindle speed	3000 r/min.

## Data Availability

The authors confirm that the data supporting the findings of this study are available within the article.

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
