# Peer review of "Experimental Study on Two-Dimensional Rotatory Ultrasonic Combined Electrochemical Generating Machining of Ceramic-Reinforced Metal Matrix Materials"

_sensors, 2022, doi:10.3390/s22030877_

Round 1
Reviewer 1 Report
This work reports the “Experimental study on two-dimensional rotatory ultrasonic composite electrochemical generating machining of ceramic reinforced metal matrix materials“. The paper is well organized and discussed in details. However, there are recent bibliography on this type of materials that it deserves to be read by the authors and addition to the reference list.
- Experimental study on properties of Al–Al2O3 nanocomposite hybridized by graphene nanosheets. Journal of Materials Research and Technology, 2020.
- Effect of Al2O3 particles on mechanical and tribological properties of Al–Mg dual-matrix nanocomposites. Ceramics International, 2020.
- Effect of nano Al2O3 coated Ag addition on the corrosion resistance and electrochemical behavior of Cu-Al2O3 nanocomposites. Journal of Materials Research and Technology, 2020.
Reviewer 2 Report
The article presents two-dimensional rotatory ultrasonic composite electrochemical generating machining of ceramic reinforced metal matrix materials. There are interesting and useful results in this article. The following issues may be addressed for further review.
(1) The article mainly focuses on experimental study. I suggest the authors make deeper analysis in the data.
(2) There are too many figures in one article. Several images can be combined into one. Also, some figures may be listed as supporting information.
(3) There are grammatical errors in the whole text. The article should be polished carefully.
Reviewer 3 Report
The paper has recurring typing errors:
- In lines 19, 123, 177, 189, 191, 475, 492 after symbol ; lowercase of the next word should be used because there is no listing or numbering in the sentence. Probably the symbol ; shall be changed to coma symbol.
- Between number and unit there should be a space in lines: 283, 284, 376, and 399.
- In tables 1. and 2. NaNO3 should be written as NaNO3.
- In the same tables there is a unit um. It should be written as μm
References [17] and [18] are listed without concise connection with this work and presented results of previous experiments. References [19-21] are not even connected with text in lines 72-76, and text explains that those references are work of different author.
The figures are too small and unclear with low resolution. Figures should be 4 to 5 times increased. Scales in fig. 2 are not visible. In fig. 10. there are blue and red lines in diagram and there are not defined in the text above.
Figures 12., 13., 15. and 17. cannot provide conclusions with such low resolution.
The English language should be checked and some sentences should be written simpler. For example in line 172 "and the etched material is more" could be changed into "there is more etched material". In line 196 "the surface quality of the workpiece is not good" should be changed into "the surface quality of the workpiece is unacceptable" . In line 274 "Figure 6. is" should be changed with "Figure 6. presents". In line 332. "When the system is machined" shlould be changed with "When the system works" or "When the system is machining". In lines 342, 350 "to judge" could be replaced with "to evaluate". In line 414 part "when machining gap is smaller and smaller" could be written as "when machining gap is to small or insufficient"
Parameters for each phase of the experiment (experiment states) could be presented with table and resulted specimens should be named.
Surface roughness should be measured on each specimen. This would exclude subjective assessment based on visual inspection of the specimens (figures 12.,13.,15. and 17.)
There is lack of values measured in the experiment in the conclusion part of this paper.
At the end of conclusion lines from 498 to 512 must be deleted because this part is instruction from the journal template.
Round 2
Reviewer 2 Report
The article has been revised according to the comments. It can now be accepted for publication.
Reviewer 3 Report
Everything has been changed well. All coments have been adopted.